# The Impact of Excitation Periods on the Outcome of Lock-In Thermography

**DOI:** 10.3390/ma16072763

**Published:** 2023-03-30

**Authors:** Milan Sapieta, Vladimír Dekýš, Peter Kopas, Lenka Jakubovičová, Zdenko Šavrnoch

**Affiliations:** Faculty of Mechanical Engineering, University of Zilina, Univerzitná 8215/1, 010 26 Žilina, Slovakia

**Keywords:** infrared camera, non-destructive testing (NDT), image processing, structural similarity index measure (*SSIM*), lock-in thermography

## Abstract

Thermal imaging is a non-destructive test method that uses an external energy source, such as a halogen lamp or flash lamp, to excite the material under test and measure the resulting temperature distribution. One of the important parameters of lock-in thermography is the number of excitation periods, which is used to calculate a phase image that shows defects or inhomogeneities in the material. The results for multiple periods can be averaged, which leads to noise suppression, but the use of a larger number of periods may cause an increase in noise due to unsynchronization of the camera and the external excitation source or may lead to heating and subsequent damage to the sample. The phase image is the most common way of representing the results of lock-in thermography, but amplitude images and complex images can also be obtained. In this study, eight measurements were performed on different samples using a thermal pulse source (flash lamp and halogen lamp) with a period of 120 s. For each sample, five phase images were calculated using different number of periods, preferably one to five periods. The phase image calculated from one period was used as a reference. To determine the effect of the number of excitation periods on the phase image, the reference phase image for one period was compared with the phase images calculated using multiple periods using the structural similarity index (*SSIM*) and multi-scale SSIM (*MS-SSIM*).

## 1. Introduction

Infrared thermography is a powerful and widely used method of non-destructive testing (NDT) and evaluation that utilizes an external energy source to create a temperature difference between areas of a sample that are defective and those that are not, in order to identify any defects. This method is known as active thermography, as opposed to passive thermography, in which defects are identified through naturally occurring thermal contrasts.

One of the key advantages of infrared thermography is that it allows for the detection of defects that may not be visible to the naked eye, such as those found inside a material or component. Additionally, it can be used to detect defects in a wide range of materials and industries, including aerospace, automotive, and electrical engineering. As a non-destructive testing method, Infrared thermography can be used to identify defects without causing any damage to the sample, making it a cost-effective and efficient method of testing. 

Lock-in thermography is a popular type of non-destructive testing that uses an external source to induce temperature changes in a material or component, and then uses an infrared camera to measure the resulting thermal response. Lock-in thermography is often used to inspect composite materials such as aerospace structural elements [1,2,3,4,5], wind turbine blade parts [6,7,8], and other advanced composites such as composites containing nanotubes [9,10] as well as sandwich structures [11]. Lock-in thermography is widely used for detecting subsurface defects, such as cracks [12,13], delamination [14,15,16], impact damage [17,18], and corrosion [19,20]. It is considered a highly versatile NDT method because it can be used for a wide range of materials and components such as metal [21], composites [22], wood [23], ceramics [24], plastics [25], and concrete [26]. It can be used in laboratory and in-field settings, and it can be used to inspect both single samples or large areas.

## 2. Lock-In Method

The lock-in principle is the technique of choice, if signals have to be extracted from statistical noise. Prerequisite to using this technique is that the primary signal, can be periodically pulsed or amplitude-modulated with a certain frequency called “lock-in frequency” [27].

It uses a reference signal, known as a local oscillator that is synchronized with the signal of interest and is used to “lock-in” or extract the small signal from the background noise. This can be done by multiplying the input signal with the reference signal and then applying a low-pass filter to the product. The result is a signal that is proportional to the original small signal, but with much-reduced noise. This method is commonly used in experimental physics, chemistry, and other scientific fields.

The concept of this method is that the response of the system, when stimulated by a signal with a sinusoidal character and lock-in frequency $f_{L}$, will also be in the form of a sinusoid with the same frequency and with amplitude A anf phase shift φ. This means that the response *s*(*t*) can be represented as a function of this sinusoidal character in the time *t*:
(1)$$s\left(t\right)=Asin\;\left(2\pi f_{L}t+\varphi \right)=acos\;\left(2\pi f_{L}t\right)+bsin\left(2\pi f_{L}t\right),$$
(2)a=sin (φ), b=cos(φ).

Given that the amplitude of the response *s*(t) is very small, and the signal S(t) being measured also contains a much higher level of noise N(t), the relationship between the two can be described as:(3)S(t)=s(t)+N(t).

If S(t) is expressed by Fourier’s series than:(4)$$S\left(t\right)=\frac{A_{0}}{2}+\overset{\infty }{\underset{i=1}{\sum }}A_{i}sin\;\left(2\pi if_{L}t+\varphi_{i}\right)=\;=A_{1}sin\left(2\pi f_{L}+\varphi_{1}\right)+\left[\frac{A_{0}}{2}+\overset{\infty }{\underset{i=2}{\sum }}A_{i}sin\left(2\pi if_{L}t+\varphi_{i}\right)\right]=A_{1}\left(2\pi f_{L}t+\varphi_{1}\right)+N\left(t\right).\;$$

If $A=A_{1}$ and $\varphi =\varphi_{1}$*,* then the coefficients a, b in Equation (2) can be estimated by using the relationships of discrete Fourier transformation:(5)$$a=\overset{n}{\underset{k=1}{\sum }}F_{k}\;cos\left(2\pi \frac{k}{n}f_{L}\right),$$
(6)$$b=\overset{n}{\underset{k=1}{\sum }}F_{k}\;sin\left(2\pi \frac{k}{n}f_{L}\right),$$
when n is number of frames recorded by infrared camera in one period, $F_{k}$ is the *k*-th frame recorded in time $t_{k}$ with sample rate of IR camera. Then the phase shift of the response *s*(t) is:(7)$$\varphi =atan\frac{a}{b}.$$

The phase image is formed by the phase shift values for each single pixel of IR camera.

## 3. Structural Similarity Index Measure (*SSIM*)

The Structural Similarity Index (*SSIM*) is a tool for measuring the similarity between two images. It was first proposed by Wang et al. in 2004 [28], to quantify the perceived quality of images and videos. The *SSIM* metric is based on a comparison of the structural information in the image, as well as the luminance and contrast of the image. It is calculated as a value between −1 and 1, with values closer to 1 indicating greater similarity between the two images. The *SSIM* metric has been widely used in image and video processing, and it has been found to correlate well with human perception of image quality. 

It was designed to improve upon traditional methods for measuring the quality of images, such as the peak signal-to-noise ratio (PSNR) and mean squared error (MSE), which rely solely on the absolute error as a metric. The *SSIM* was developed to address this limitation by considering three independent factors, luminance, contrast, and structure, in its calculation, providing a more comprehensive measure of image similarity.

The general form of the Structural Similarity (*SSIM*) Index between *x* and *y* signals is defined as [28]:(8)$$SSIM\left(x,\;y\right)=\left[l\left(x,\;y\right)\right]{\;}^{\alpha }\cdot {\left[c\left(x,\;y\right)\right]}^{\beta }\;\cdot {\left[s\left(x,\;y\right)\right]}^{\gamma }$$
(9)$$l\left(x,\;y\right)=\frac{2\mu_{x}\mu_{y}+C_{1}}{\mu_{x}^{2}+\mu_{y}^{2}+C_{1}},$$
(10)$$c\left(x,\;y\right)=\frac{2\sigma_{x}\sigma_{y}+C_{2}}{\sigma_{x}^{2}+\sigma_{y}^{2}+C_{2}},$$
(11)$$s\left(x,\;y\right)=\frac{\sigma_{xy}+C_{3}}{\sigma_{x}\sigma_{y}+C_{3}},$$
where *μ_x_*, *μ_y_* are the local means, *σ_x_*, *σ_y_* are standard deviations and *σ_xy_* is cross-covariance for images *x, y.* If *α* = *β* = *γ* = 1 (the default values for exponents), and *C*_3_* = C*_2_/2 (default value of *C*_3_) the index simplifies to [28]:(12)$$SSIM\left(x,\;y\right)=\frac{\left(2\mu_{x}\mu_{y}+C_{1}\right)\left(2\sigma_{xy}+C_{2}\right)}{\left(\mu_{x}^{2}+\mu_{y}^{2}+C_{1}\right)\left(\sigma_{x}^{2}+\sigma_{y}^{2}+C_{2}\right)},$$
which satisfies the following conditions [28]:Symmetry: *SSIM(x, y)* = *SSIM(y, x)*;Boundedness: *SSIM(x, y)* < 1;Unique maximum: *SSIM(x, y)* = 1 if and only if *x* = *y*.

### Multi-Scale SSIM Index (MS-SSIM)

Multi-scale method is a convenient way to incorporate image details at different resolutions. Experiments show that with an appropriate parameter setting, the multi-scale method outperforms the best single-scale *SSIM* model as well as state-of-the-art image quality metrics [28].

Multi-scale *SSIM* is calculated by first converting the two images to grayscale, if they are not already. Next, the images are down-sampled to multiple scales using a Gaussian pyramid. For each scale, the *SSIM* index is calculated between the two images. The final *MS-SSIM* value is obtained by averaging the *SSIM* values across all scales. Finally, the *MS-SSIM* value is a single value between −1 and 1, where 1 indicates a perfect match and −1 indicates no similarity.

## 4. Materials and Methods

### 4.1. Test Samples

Eight samples, each measuring 120 mm × 120 mm × 5 mm and containing defects, were printed using a 3D printer (Prusa i3 MK3S+, Prusa Research a.s., Partyzánská 188/7A 17000 Praha 7, Czech Republic) with PET-G (polyethylene terephthalate glycol, Table 1 [29]) material and FDM (fused deposition modeling) technology. FDM 3D printers work by extruding thermoplastic filaments through a heated nozzle, melting the material and layering it onto a build platform until the part is complete. 

The defects, in the form of blind holes, had known dimensions, geometry, and location. For this study, eight samples were made: S1, S2, S3, and S4 for the samples with circular cross-section defects, and S5, S6, S7, and S8 for the samples with square cross-section defects. Defects were placed 1 mm and 0.5 mm under the measured surface and the midpoints of the defects closest to the edges of the samples had a distance of 30 mm. The samples are shown in Figure 1, where the defects with square cross-sections have an edge length of 20 mm and the defects with circular cross-sections have a diameter of 20 mm. On the measured surface of the test samples a black color with the high emissivity (0.96) was applied to reduce the reflection of the surface.

The additively manufactured specimens were built with full density.

### 4.2. Experimental Setup

Test samples were measured using lock-in thermography. In lock-in thermography, various types of excitations are used, such as laser, halogen lamp, ultrasound sonotrode, flash lamp, etc. In this experiment, two sources of excitation were used: a halogen lamp (Kaiser Videolight 4, Kaiser Fototechnik GmbH & Co. KG, Buchen, German) and a flash lamp (Godox QS 1200 II, GODOX Photo Equipment Co.,Ltd., Shenzhen, China). Using the National Instruments module NI 9481, the halogen and flash lamps were connected to the PC and a control script was written in MATLAB. In the case of the halogen lamp, the device was turned on and off every 1 min, and in the case of the flash lamp, it was fired every 2 min. The halogen lamp controlled by the NI 9481 module was used to generate a one-minute thermal wave, and the flash lamp generated a flash with duration 1/800 s. Every period of excitation was 2 min because the lock-infrequency was set to 1/120 Hz. Five periods were performed for each measurement to investigate the effect of the excitation periods number on the results. Table 2 provides a summary of the setup parameters used for the experiment.

For collecting the data (sequences of thermograms), we used an infrared camera (FLIR SC7500, FLIR Systems Advanced Thermal Solutions SA, Croissy-Beaubourg, France) with thermal sensitivity less than 20 mK, a spectral range 1.5–5.1 µm and 50 mm lens (objective). The cooled detector is a focal plane array (FPA) with resolution of 320 × 256, made of InSb. Emissivity ε = 0.96 was adopted for the tests because the black color with the same emissivity value were placed on the samples. Figure 2 shows a schematic representation of the experimental setup. 

The defects are not located inside the specimens but on one side of the specimen so that they are located 1 mm and 0.5 mm below the surface, respectively. For the measurements, the samples were turned towards the camera so that the camera could see the side of the sample without defects, so that from the camera’s point of view the defects were below the surface.

Control signals for switching on and off the halogen lamp and also for switching on the flash lamp were generated in Matlab through the NI 9481 control unit.

### 4.3. Data Processing and Phase Image Calculation

Each measurement consisted of 3000 thermograms recorded over a 10-min period. The camera was positioned 0.5 m away from the sample’s surface being measured. The data was collected using the ResearchIR software and exported to a Matlab file. The exported data included sequences of thermograms and files containing information about the date and time each thermogram was taken. We created a script in Matlab to load this data and calculate the phase image (using Formula (7)) for one period (600 thermograms) up to five periods (3000 thermograms). The results were then compared using the *SSIM* and *MS-SSIM* indices. *SSIM* method was chosen because it provides a reliable and accurate measure of the structural similarity between two images, and *SSIM* is well-established and trusted in many image and video processing applications. The *MS-SSIM* was used as a second comparative method and because it outperforms the single-scale *SSIM* method.

## 5. Results

In this paper, two excitation methods were used: flash lamp and halogen lamp. For each method, eight measurements were performed. The samples were designed so that the defects occupied different areas on the measured surface, ranging from approximately 8% for three defects to 25% for nine defects.

Figure 3 and Figure 4 present a sample thermogram from the recorded sequences of thermograms on the left side. The mean temperature value for the highlighted area is plotted on the graph labeled ”Output” located on the right side. The graph labeled ”Input” shows the input signal The signal was generated in Matlab and used to control the flash lamp and halogen lamp through the NI 9481 control unit.

### 5.1. Lock-In Thermography Using Flash Lamp Defects Placed 1 mm under the Measured Surface

Figure 5 and Figure 6 display phase images calculated for all eight samples for one to five periods. The reference phase images were calculated using data from a 2-min record, whereas the phase image for five periods was calculated using data from a 10-min record.

The phase images calculated for different periods of time, ranging from one to five, are very similar in appearance. To measure the similarity between these images, the *SSIM* index and *MS-SSIM* index were used. The reference phase images, calculated from one period of time, were used as the benchmark to compare against the other phase images calculated for different periods of time. The results, as shown in Table 3, demonstrate that the difference between the reference phase images and the other phase images calculated for different periods of time is quite small. In fact, the difference, as measured by the *SSIM* index, is less than 10% and, when measured by the *MS-SSIM* index, is less than 5%. This indicates a high level of similarity between the phase images calculated for different periods of time, providing confidence in the accuracy and reliability of the calculations performed.

The Figure 7 shows graphically the *SSIM* and *MS-SSIM* index values for the flash lamp. A decrease in the index values with increasing number of periods was observed. We assume that the extreme values of the decrease are due to the increasing noise in the processed data.

### 5.2. Lock-In Thermography Using Halogen Lamp Defects Placed 1 mm under the Measured Surface

The data processing and phase image calculation procedures are similar to those used in the case of lock-in thermography using a flash lamp, as described in Section 4.1. 

The Figure 8 shows graphically the *SSIM* and *MS-SSIM* index values for a halogen lamp. A decrease in the index values with increasing number of periods was observed. We assume that the extreme values of the decrease are due to the increasing noise in the processed data.

Figure 9 and Figure 10 present phase images calculated for all eight samples for periods ranging from one to five. It can be observed from the figures that the phase images are similar, but not as clear as in the previous case. With an increase in the number of periods, the phase images become more blurred, as seen in the comparison between the S3 reference and S3 for four periods. Results in Table 4 indicate that, using both the *SSIM* and *MS-SSIM* indices, the difference between the reference phase images and phase images calculated for two periods is less than 10%. However, as the number of periods increases, the percentage differences also increase.

### 5.3. Lock-In Thermography Using Halogen Lamp and Flash Lamp. Defects Placed 0.5 mm under the Measured Surface

The next measurement that was performed was on samples that had a defect depth of 0.5 mm below the surface of the measured area. The sample was extruded from PET-G material. In this case, a sample with eight square holes, also marked as S8, is chosen. The measurement setup is the same as for the 1 mm thickness samples. The measurement was performed for both the flash lamp and the halogen lamp excitation types. The phase image from the halogen lamp excitation can be observed in Figure 11 and the phase image from the flash lamp excitation can be observed in Figure 12. The representations are made for five periods. The comparison of the *SSIM* and *MS-SSIM* indexes is shown in Table 5.

The graph indicating the dependence of the *MS-SSIM* and *SSIM* indices on the number of periods is on the left side for flash lamp excitation and on the right side for halogen lamp excitation. The graphs are made for the comparison of sample S8 wall thickness of the measured area of 1 mm and 0.5 mm (Figure 13).

## 6. Discussion

The case of excitation by flash lamp. Increasing the number of periods used in the lock-in processing does not increase the value of the *MS-SSIM* index or the *SSIM* for the 1 mm and 0.5 samples This observation is represented by the triangular points in Figure 13 for the flash lamp.

We consider the slower decline of the *MS-SSIM* index relative to the *SSIM* index to be due to the lower sensitivity of the *MS-SSIM* to period growth.

The case of excitation by halogen lamp. Increasing the number of periods used in the lock-in data processing does not significantly increase the value of the *MS-SSIM* index or the *SSIM* index for the 1 mm sample, and the index values shown are lower for higher numbers of periods. However, in the case of a 0.5 mm sample, increasing the number of periods used in the data processing by the lock-in method increases significantly the value of the *MS-SSIM* index and not the *SSIM* for the 0.5 mm sample. In this case, we cannot claim that one period is sufficient.

The presented results for the halogen lamp excitation indicate a possible dependence of the *MS-SSIM* and *SSIM* index values on the depth below the surface in which the defect is located. We assume that the halogen lamp heated the sample more than the flash lamp, and then the increase of the indices at 0.5 mm sample may be due to this phenomenon. However, the rising temperature when measuring using a halogen lamp may cause sample damage if the sample is excited for a large number of periods.

## 7. Conclusions

In this paper, two excitation methods, namely flash lamp and halogen lamp, were used to perform eight measurements on samples. The phase images were calculated for periods ranging from one to five, and the similarity index was calculated between the reference phase image and others. The data processing and phase image calculation procedures were similar to those used in the case of lock-in thermography using a flash lamp.

For the measurements on the 1 mm and 0.5 mm depth of the defect surfaces, the same lamp power was set for both the halogen lamp and the flash lamp, and the flash power was also set to be the same. The same lock-in periods, camera frame rates, and camera integration times were also used, and the same distance of the samples from the excitation sources was maintained.

Using a flash lamp, it was found that increasing the number of periods that were used in the data processing by the lock-in method did not increase the value of the *MS-SSIM* index or the *SSIM* for the 1 mm sample. For the 0.5 mm sample, the increase in value is not demonstrable.

If we subjectively or visually evaluate (0.5 mm sample) the images for one to five periods (see Figure 13), then the image for five periods appear blurrier than for one period, although the values of both indices increase with increasing number of periods. 

Based on the above, in future research we will want to observe the influence of other parameters on the value of the above indices as well as the influence of other parameters, such as the power of the halogen reflector or the temperature of the sample during the test, and to make these measurements for other defect depths below the surface as well.

In the case of excitation of an object using a flash lamp, we recommend to use only one period, and we cannot make a recommendation in the case of using a halogen lamp.

## Figures and Tables

**Figure 1 materials-16-02763-f001:**
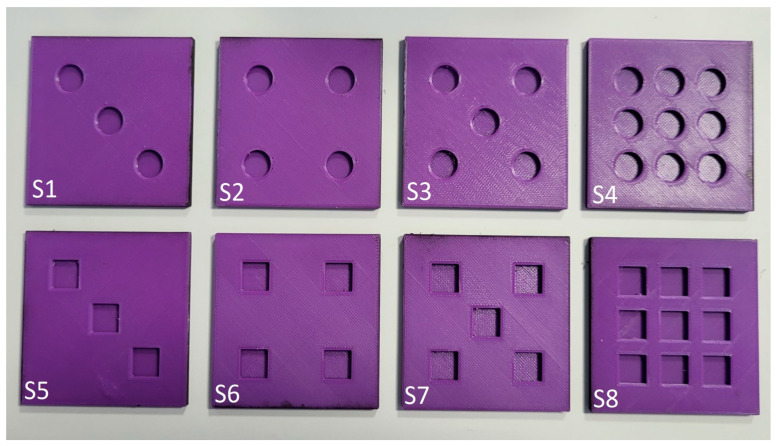
Test samples.

**Figure 2 materials-16-02763-f002:**
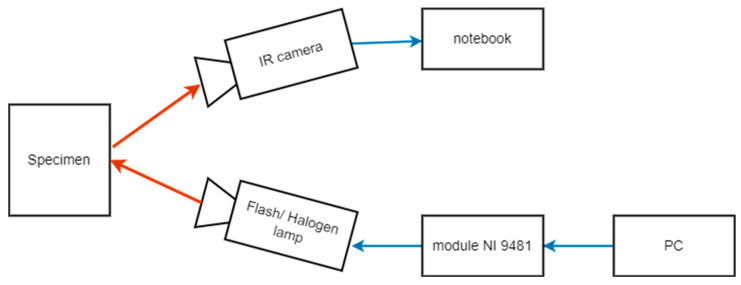
Experimental configuration of a lock-in thermography setup.

**Figure 3 materials-16-02763-f003:**
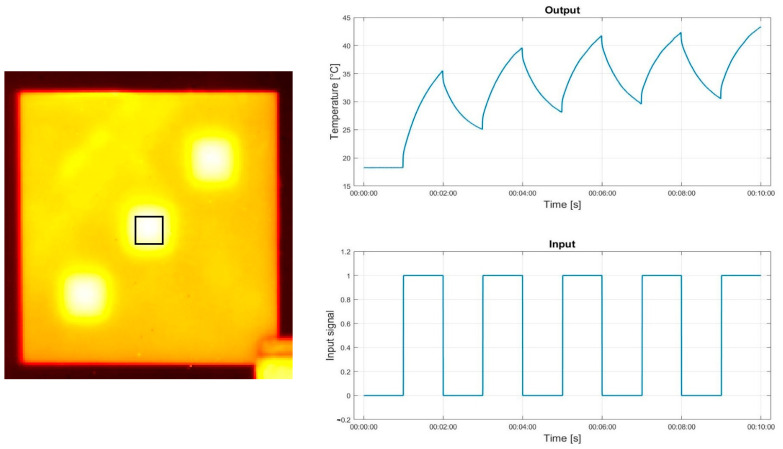
Halogen lamp input and output signal.

**Figure 4 materials-16-02763-f004:**
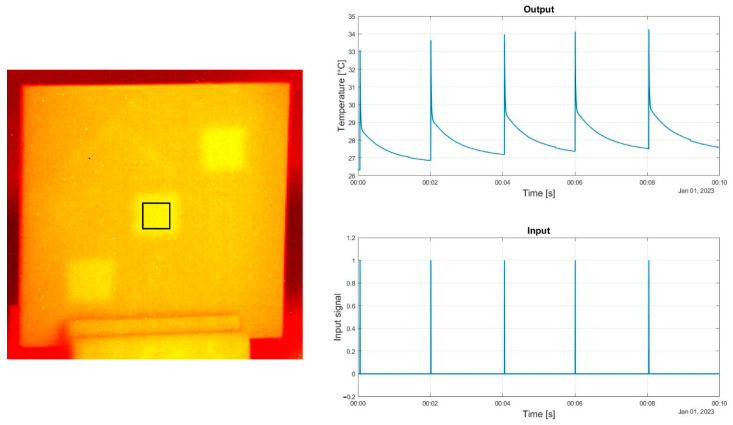
Flash lamp input and output signal.

**Figure 5 materials-16-02763-f005:**
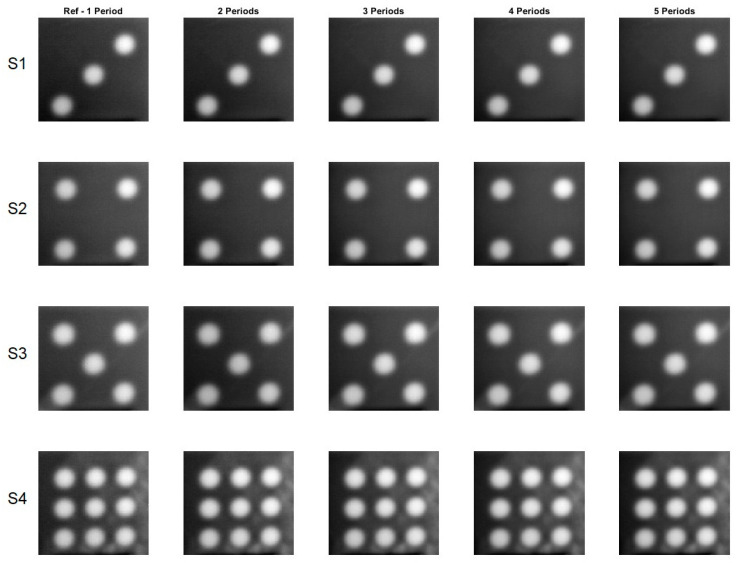
Phase images for the samples with the circular cross section defects. A flash lamp was used as a source of excitation. S1–S4 denotes configuration of the circle holes.

**Figure 6 materials-16-02763-f006:**
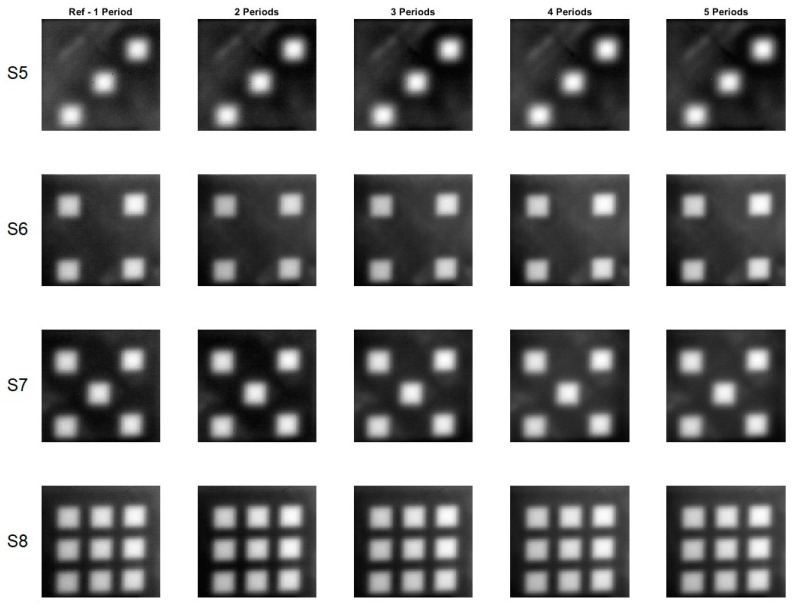
Phase images for the samples with the square cross section defects. A flash lamp was used as a source of excitation. S5–S8 denotes configuration of the square holes.

**Figure 7 materials-16-02763-f007:**
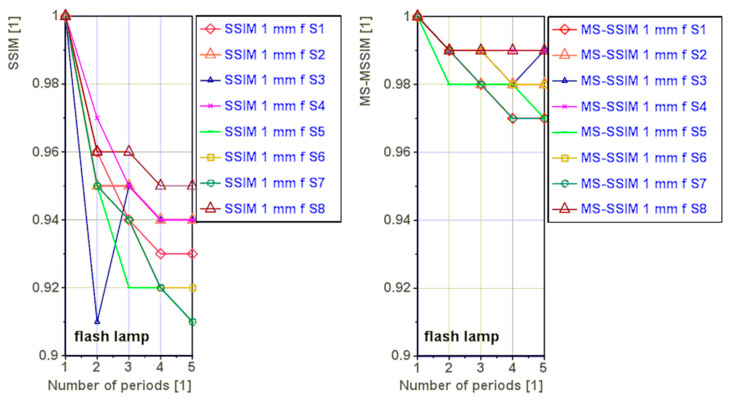
Variation of the *SSIM* and *MS-SSIM* index values as a function of the number of periods for the flash lamp.

**Figure 8 materials-16-02763-f008:**
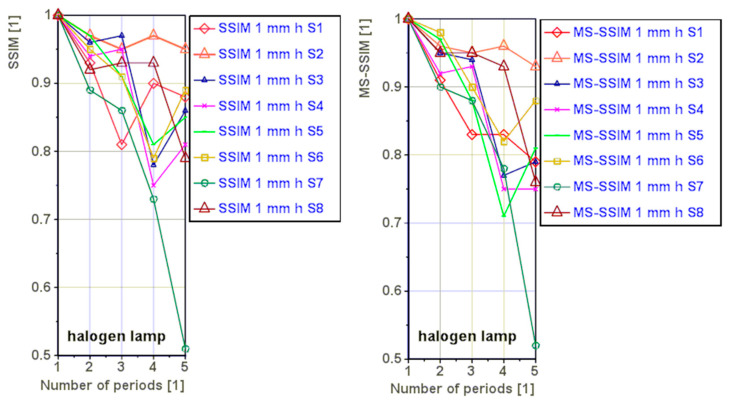
Variation of the *SSIM* and *MS-SSIM* index values as a function of the number of periods for the halogen lamp.

**Figure 9 materials-16-02763-f009:**
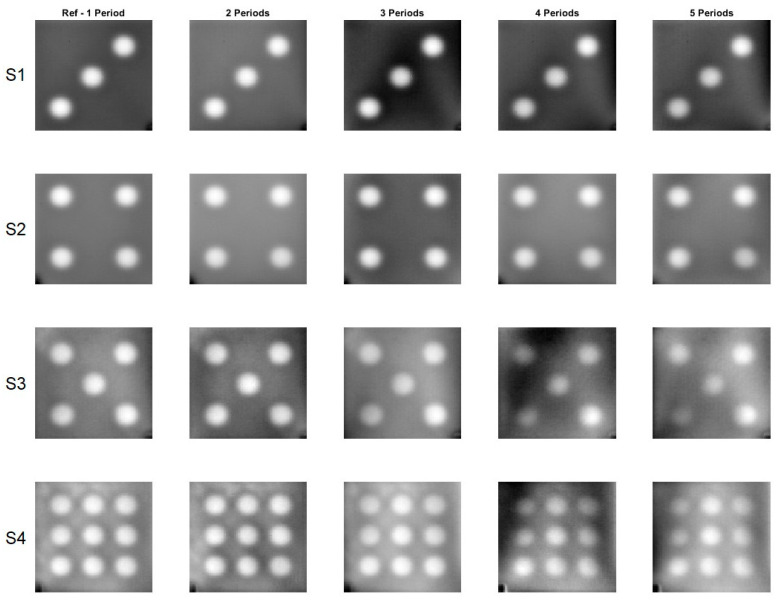
Phase images for the samples with the circular cross section defects. A halogen lamp was used as a source of excitation. S1–S4 denotes configuration of the circle holes.

**Figure 10 materials-16-02763-f010:**
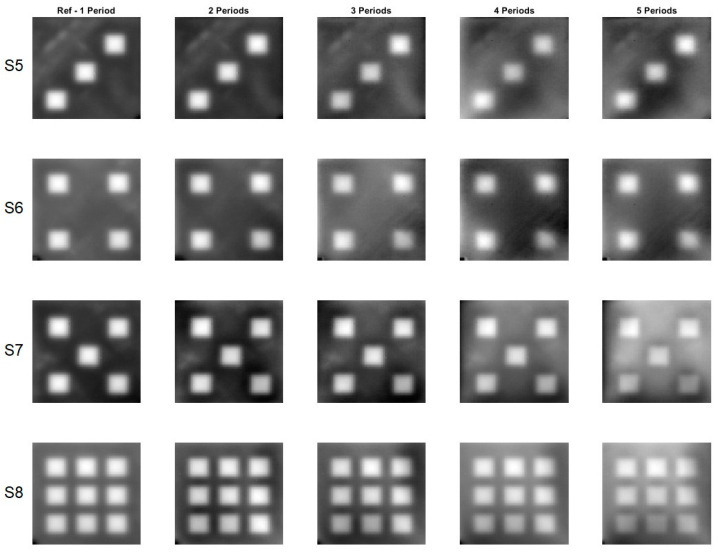
Phase images for the samples with the square cross section defects. A halogen lamp was used as a source of excitation. S5–S8 denotes configuration of the square holes.

**Figure 11 materials-16-02763-f011:**

Phase images for the samples with the square cross section defects. A halogen lamp was used as a source of excitation. Defects were placed 0.5 mm under the measured surface.

**Figure 12 materials-16-02763-f012:**

Phase images for the samples with the square cross section defects. A flash lamp was used as a source of excitation. Defects were placed 0.5 mm under the measured surface.

**Figure 13 materials-16-02763-f013:**
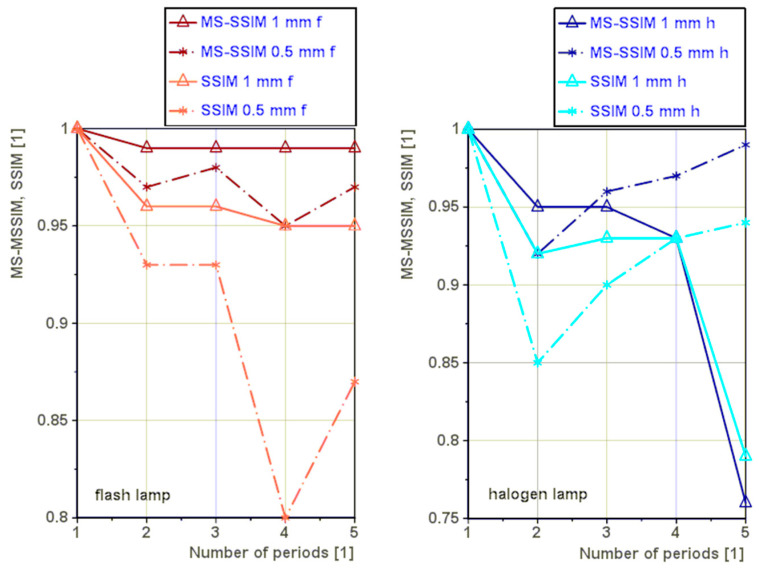
Comparison of the dependence of the *MS-SSIM* and *SSIM* indices on the number of periods on the left side for flash lamp excitation and on the right side for halogen lamp excitation.

**Table 1 materials-16-02763-t001:** PETG, the selected parameters of mechanical and thermal properties (injection molded), ISO method, [29].

Parameter	Test Method	Value
Density	ISO 1183, Method D	1.27 g/cm^3^
Tensile stress @ break	ISO 527	28 MPa
Tensile stress @ yield	ISO 527	50 MPa
Elongation @ break	ISO 527	100%
Tensile modulus	ISO 527	2100 MPa
Deflection temperature		
@ 0.455 MPa	D 648	70 °C
@ 1.82 MPa	D 648	64 °C
Vicat softening temperature	D 1525	85 °C
Glass transition temperature (Tg)	DSC	80 °C
Coefficient of linear thermal expansion (−30 °C to 40 °C)	D 696	5.1 × 10^−5^/°C (mm/mm·°C)

**Table 2 materials-16-02763-t002:** Specification of the parameters used for lock-in thermography.

Parameter	Value
Distance sample to IR camera	0.5 m ± 0.1 m
Distance sample to halogen lamp	0.5 m ± 0.1 m
Distance sample to flash lamp	0.5 m ± 0.1 m
IR camera frame rate	5 Hz
Lock-in frequency	1/120 Hz
Number of periods	5
Power of halogen lamp	1 kW
Power of flash lamp	1.2 kW

**Table 3 materials-16-02763-t003:** Comparation of phase images for all measurement in the case of a flash lamp as source of excitation.

Sample	*SSIM* Index *MS-SSIM* Index			
	2 Periods	3 Periods	4 Periods	5 Periods
S1	0.96 0.99	0.94 0.98	0.93 0.97	0.93 0.97
S2	0.95 0.99	0.95 0.98	0.94 0.98	0.94 0.98
S3	0.91 0.99	0.95 0.99	0.94 0.98	0.94 0.99
S4	0.97 0.99	0.95 0.99	0.94 0.99	0.94 0.99
S5	0.95 0.98	0.92 0.98	0.92 0.98	0.91 0.97
S6	0.95 0.99	0.94 0.99	0.92 0.98	0.92 0.98
S7	0.95 0.99	0.94 0.98	0.92 0.97	0.91 0.97
S8	0.96 0.99	0.96 0.99	0.95 0.99	0.95 0.99

**Table 4 materials-16-02763-t004:** Comparation of phase images for all measurement in the case of halogen lamp as source of excitation.

Sample	*SSIM* Index *MS-SSIM* Index			
	2 Periods	3 Periods	4 Periods	5 Periods
S1	0.93 0.91	0.81 0.83	0.90 0.83	0.88 0.79
S2	0.97 0.96	0.95 0.95	0.97 0.96	0.95 0.93
S3	0.96 0.95	0.97 0.94	0.78 0.77	0.86 0.79
S4	0.94 0.92	0.95 0.93	0.75 0.75	0.81 0.75
S5	0.97 0.97	0.91 0.88	0.81 0.71	0.85 0.81
S6	0.95 0.98	0.91 0.90	0.79 0.82	0.89 0.88
S7	0.89 0.90	0.86 0.88	0.73 0.78	0.51 0.52
S8	0.92 0.95	0.93 0.95	0.93 0.93	0.79 0.76

**Table 5 materials-16-02763-t005:** Comparation of phase images for measurement in the case of the halogen lamp and the flash lamp as source of excitation for defects place 0.5 mm under measured surface.

Excitation	*SSIM* Index *MS-SSIM* Index			
	2 Periods	3 Periods	4 Periods	5 Periods
Flash lamp	0.93 0.97	0.93 0.98	0.8 0.95	0.87 0.97
Halogen lamp	0. 85 0. 92	0.9 0.96	0.93 0.97	0.94 0.99

## Data Availability

The data that support the findings of this study are available from the corresponding author upon request.

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
