# Peer review of "The Impact of Excitation Periods on the Outcome of Lock-In Thermography"

_materials, 2023, doi:10.3390/ma16072763_

Round 1
Reviewer 1 Report
The manuscript focuses on the excitation period effect on the outcome of lock-in thermography. The authors propose a comparison between two heating sources, considering a variation of the defect number. However, I would improve the outcomes of the work with wider considerations. Therefore, in my opinion the work can be accepted after major revisions, which are listed in the following.
The authors considered the effect of the defect number. However, they are all located at the same depth respect to the measured surface. I would enrich the analysis adding few samples with defect located at least to a second depth from the measured surface. The study on the optimum excitation period could be conducted as function of this further variable.
In the “Materials and Methods” chapter the authors did not mention if the additively manufactured specimens were built with a full dense theme of if a reduction of density was applied to fill the layers.
Figure 3 should be moved to the “Results” chapter because the temperature profile is an outcome of the experiments. I suggest to provide this figure for all the selected depths, possibly selecting defects located in the same position.
In the manuscript the “Discussion” chapter is missing.
Author Response
We thank the reviewer for their supportive comments and the time they spend reviewing our submission. We very much appreciate their efforts to improve the submission. Next, we explain the changes we made to the manuscript based on the received comments. All the changes in the revised submission are marked with red.
The manuscript focuses on the excitation period effect on the outcome of lock-in thermography. The authors propose a comparison between two heating sources, considering a variation of the defect number. However, I would improve the outcomes of the work with wider considerations. Therefore, in my opinion the work can be accepted after major revisions, which are listed in the following.
The authors considered the effect of the defect number. However, they are all located at the same depth respect to the measured surface. I would enrich the analysis adding few samples with defect located at least to a second depth from the measured surface. The study on the optimum excitation period could be conducted as function of this further variable.
Measurements were taken on a sample with a thickness of the surface of 0.5 mm.
In the “Materials and Methods” chapter the authors did not mention if the additively manufactured specimens were built with a full dense theme of if a reduction of density was applied to fill the layers.
Specimens were built with a full dense.
Figure 3 should be moved to the “Results” chapter because the temperature profile is an outcome of the experiments. I suggest to provide this figure for all the selected depths, possibly selecting defects located in the same position.
Figure 3 has been moved to the Results.
In the manuscript the “Discussion” chapter is missing.
Discussion chapter was added.

Reviewer 2 Report
1. The subject addressed in the article is of current interest and consistent with the journal's profile (research conducted on test samples of plastic material).
2. The research results were not sufficiently highlighted in the abstract.
3. All symbols entered in the equations must be explained. For example, for the symbols ϕ, t in equation (1), but also other places (for example, for the symbols included in equation (7)), there are no adequate explanations in the text of the article.
4. More information about the PET-G plastic material could be included in the article, possibly with the formulation of some hypotheses regarding the possibility that some properties of this material may influence the results of the experimental research.
4. The obtained experimental results could be used to create graphic representations, identify empirical mathematical models, etc.
5. The defects do not seem to be inside the samples ("All defects were placed 1 mm under the measured surface") but on one of the large flat surfaces of the test sample.
6. The units of measurement (kW.s) for the power of the flash lamp in Table 1 do not seem to be adequate for rating the power. However, it is recommended to have a space between the number and the units of measure.
7. Some quantitative evaluations and comparisons of the experimental results could be introduced in the results and conclusions chapters, respectively.
8. A wording that could suggest possibilities for further research in the future could be inserted at the end of the conclusions chapter.
9. Authors should pay more attention to article editing and English expression.
Thus, there are confusing expressions. For example:
- "Lock-in thermography is often used to inspect composite materials, such as aircraft and aerospace structures [1-5], wind turbine blades [6-8], and other advanced composites like nanotubes [9,10] and sandwich structures [11]”. The aircraft and aerospace structures, wind turbine blades are not, in themselves, composite materials, just as nanotubes are not composite materials either;
- "Structural Similarity Index (SSIM) is a method for measuring ..." is confusing. A similarity index cannot be a method;
- "and the similarity was calculated" in the first paragraph of the conclusions. A similarity index and not similarity was calculated.
There are instances where uppercase letters were mistakenly used instead of lowercase letters. E.g., "testing method, Infrared thermography" in the second paragraph of the Introduction, "the relationships of Discrete Fourier transformation:" in the paragraph preceding relationship (5), "as the Peak signal-to-noise ratio (PSNR) and Mean squared error” in the first paragraph of page 3, ” Indium Antimonide” in the first paragraph of page 5, ”in the case of Lock-in thermography” in the first paragraph of subsection 4.2, ”the default values for Exponents, in the paragraph after the equation (11), etc.
The explanation of some symbols written in equations starts from the left end of the line, without using the TAB function in Word (for example, after equations (6), (7), (11), etc.).
At the end of the first paragraphs of subchapter 3.1 and chapter 2, full stops should be placed after the references and not before the references to those references.
A full stop is not necessary after the titles of subchapters 4.2 and 4.3.
All symbols of different sizes should be written using italics (e.g., the symbols µx, µy, x, y, etc., on page 3). No italics shall be used for writing equation numbers.
The formulation "Which satisfies the following" found after equation (12) could start with a lowercase letter, and a comma could be placed after equation (12).
The punctuation mark semicolon could be used at the end of the first two components of the enumeration located after equation (12). At the same time, the first words of the enumeration components could start with capital letters (in English, after a full stop, it always uses a capital letter).
The abbreviated forms of the journal titles were not used in the bibliographic reference list.
There is an inconsistency in the writing of the titles of some articles included in the bibliographic reference list (in the case of some article titles, the main concepts were written in capital letters at the beginning of the words - see references [1], [2], [3], etc., while in other cases, only lowercase letters were used, except for the first letter of the first word of the article title – see references [4], [5], [6], etc.).
It is unnecessary to use bold characters when writing some symbols (for example, in the text on page 2), but only in italic fonts, as previously mentioned.
It can write "index between the x and y signals" instead of "index between signal x and y".
Author Response
We thank the reviewer for their supportive comments and the time they spend reviewing our submission. We very much appreciate their efforts to improve the submission. Next, we explain the changes we made to the manuscript based on the received comments. All the changes in the revised submission are marked with red.
- The subject addressed in the article is of current interest and consistent with the journal's profile (research conducted on test samples of plastic material).
- The research results were not sufficiently highlighted in the abstract.
Results were highlighted in the abstract.
All symbols entered in the equations must be explained. For example, for the symbolsϕ,t in equation (1), but also other places (for example, for the symbols included in equation (7)), there are no adequate explanations in the text of the article.
All symbols entered in the equations have been explained.
More information about the PET-G plastic material could be included in the article, possibly with the formulation of some hypotheses regarding the possibility that some properties of this material may influence the results of the experimental research.
Information about the PET-G plastic material added to article.
The obtained experimental results could be used to create graphic representations, identify empirical mathematical models, etc.
Graphic representations were created.
The defects do not seem to be inside the samples ("All defects were placed 1 mm under the measured surface") but on one of the large flat surfaces of the test sample.
This has now been corrected.
The units of measurement (kW.s) for the power of the flash lamp in Table 1 do not seem to be adequate for rating the power. However, it is recommended to have a space between the number and the units of measure.
This has now been corrected.
Some quantitative evaluations and comparisons of the experimental results could be introduced in the results and conclusions chapters, respectively.
Quantitative evaluations and comparisons have been added in results and conclusion.
A wording that could suggest possibilities for further research in the future could be inserted at the end of the conclusions chapter.
Possibilities for further research have been added.
- Authors should pay more attention to article editing and English expression.
Thus, there are confusing expressions. For example:
"Lock-in thermography is often used to inspect composite materials, such as aircraft and aerospace structures [1-5], wind turbine blades [6-8], and other advanced composites like nanotubes [9,10] and sandwich structures [11]”. The aircraft and aerospace structures, wind turbine blades are not, in themselves, composite materials, just as nanotubes are not composite materials either;
This has now been corrected.
"Structural Similarity Index (SSIM) is a method for measuring ..." is confusing. A similarity index cannot be a method;
This has now been corrected.
"and the similarity was calculated" in the first paragraph of the conclusions. A similarity index and not similarity was calculated.
This has now been corrected.
There are instances where uppercase letters were mistakenly used instead of lowercase letters. E.g., "testing method, Infrared thermography" in the second paragraph of the Introduction, "the relationships of Discrete Fourier transformation:" in the paragraph preceding relationship (5), "as the Peak signal-to-noise ratio (PSNR) and Mean squared error” in the first paragraph of page 3, ” Indium Antimonide” in the first paragraph of page 5, ”in the case of Lock-in thermography” in the first paragraph of subsection 4.2, ”the default values for Exponents, in the paragraph after the equation (11), etc.
This has now been corrected.
The explanation of some symbols written in equations starts from the left end of the line, without using the TAB function in Word (for example, after equations (6), (7), (11), etc.).
This has now been corrected.
At the end of the first paragraphs of subchapter 3.1 and chapter 2, full stops should be placed after the references and not before the references to those references.
This has now been corrected.
A full stop is not necessary after the titles of subchapters 4.2 and 4.3.
This has now been corrected.
All symbols of different sizes should be written using italics (e.g., the symbols µx, µy, x, y, etc., on page 3). No italics shall be used for writing equation numbers.
This has now been corrected.
The formulation "Which satisfies the following" found after equation (12) could start with a lowercase letter, and a comma could be placed after equation (12).
This has now been corrected.
The punctuation mark semicolon could be used at the end of the first two components of the enumeration located after equation (12). At the same time, the first words of the enumeration components could start with capital letters (in English, after a full stop, it always uses a capital letter).
This has now been corrected.
The abbreviated forms of the journal titles were not used in the bibliographic reference list.
This has now been corrected.
There is an inconsistency in the writing of the titles of some articles included in the bibliographic reference list (in the case of some article titles, the main concepts were written in capital letters at the beginning of the words - see references [1], [2], [3], etc., while in other cases, only lowercase letters were used, except for the first letter of the first word of the article title – see references [4], [5], [6], etc.).
This has now been corrected.
It is unnecessary to use bold characters when writing some symbols (for example, in the text on page 2), but only in italic fonts, as previously mentioned.
This has now been corrected.
It can write "index between the x and y signals" instead of "index between signal x and y".
This has now been corrected.

Round 2
Reviewer 1 Report
The authors implemented the required revisions and finalized a good manuscript. In my opinion, the work can be accepted in the present form.